# Genetics of Plasma Bilirubin and Associations between Bilirubin and Cardiometabolic Risk Profiles in Danish Children and Adolescents

**DOI:** 10.3390/antiox12081613

**Published:** 2023-08-15

**Authors:** Asmat Ullah, Evelina Stankevic, Louise Aas Holm, Sara E. Stinson, Helene Bæk Juel, Cilius E. Fonvig, Morten A. V. Lund, Cæcilie Trier, Line Engelbrechtsen, Lars Ängquist, Anna E. Jonsson, Oluf Pedersen, Niels Grarup, Jens-Christian Holm, Torben Hansen

**Affiliations:** 1Novo Nordisk Foundation Center for Basic Metabolic Research, Faculty of Health and Medical Sciences, University of Copenhagen, 2200 Copenhagen, Denmark or asmatullah@bs.qau.edu.pk (A.U.); evelina@sund.ku.dk (E.S.); louise.aas@sund.ku.dk (L.A.H.); sara.stinson@sund.ku.dk (S.E.S.); helene.baek.juel@sund.ku.dk (H.B.J.); crfo@regionsjaelland.dk (C.E.F.); line@sund.ku.dk (L.E.); lars.angquist@sund.ku.dk (L.Ä.); jonsson@sund.ku.dk (A.E.J.); oluf@sund.ku.dk (O.P.); niels.grarup@sund.ku.dk (N.G.); 2The Children’s Obesity Clinic, Accredited European Centre for Obesity Management, Department of Pediatrics, Holbæk Hospital, 4300 Holbæk, Denmark; morten.lund@sund.ku.dk (M.A.V.L.); cats@regionsjaelland.dk (C.T.); 3The Faculty of Health and Medical Sciences, University of Copenhagen, 2200 Copenhagen, Denmark; 4Clinical Center for Metabolic Research, Herlev-Gentofte University Hospital, 2900 Copenhagen, Denmark

**Keywords:** bilirubin, cardiometabolic risk factors, inflammatory cytokines, GWAS, UGT1A1

## Abstract

Bilirubin is the end product of heme catabolism, mainly produced by the breakdown of mature red blood cells. Due to its anti-inflammatory, antioxidant, antidiabetic, and antilipemic properties, circulating bilirubin concentrations are inversely associated with the risk of cardiovascular disease, type 2 diabetes, and all-cause mortality in adults. Some genetic loci associated with circulating bilirubin concentrations have been identified by genome-wide association studies in adults. We aimed to examine the relationship between circulating bilirubin, cardiometabolic risk factors, and inflammation in children and adolescents and the genetic architecture of plasma bilirubin concentrations. We measured fasting plasma bilirubin, cardiometabolic risk factors, and inflammatory markers in a sample of Danish children and adolescents with overweight or obesity (*n* = 1530) and in a population-based sample (*n* = 1820) of Danish children and adolescents. Linear and logistic regression analyses were performed to analyze the associations between bilirubin, cardiometabolic risk factors, and inflammatory markers. A genome-wide association study (GWAS) of fasting plasma concentrations of bilirubin was performed in children and adolescents with overweight or obesity and in a population-based sample. Bilirubin is associated inversely and significantly with a number of cardiometabolic risk factors, including body mass index (BMI) standard deviation scores (SDS), waist circumference, high-sensitivity C-reactive protein (hs-CRP), homeostatic model assessment for insulin resistance (HOMA-IR), hemoglobin A1c (HbA1c), low-density lipoprotein cholesterol (LDL-C), triglycerides, and the majority of measured inflammatory markers. In contrast, bilirubin was positively associated with fasting plasma concentrations of alanine transaminase (ALT), high-density lipoprotein cholesterol (HDL-C), systolic blood pressure (SDS), and the inflammatory markers GH, PTX3, THBS2, TNFRSF9, PGF, PAPPA, GT, CCL23, CX3CL1, SCF, and TRANCE. The GWAS showed that two loci were positively associated with plasma bilirubin concentrations at a *p*-value threshold of <5 × 10^−8^ (rs76999922: β = −0.65 SD; *p* = 4.3 × 10^−8^, and rs887829: β = 0.78 SD; *p* = 2.9 × 10^−247^). Approximately 25% of the variance in plasma bilirubin concentration was explained by rs887829. The rs887829 was not significantly associated with any of the mentioned cardiometabolic risk factors except for hs-CRP. Our findings suggest that plasma concentrations of bilirubin non-causally associates with cardiometabolic risk factors in children and adolescents.

## 1. Introduction

Bilirubin is the end product of heme catabolism in aging red blood cells and acts as an antioxidant on low-density lipoproteins, among others [1,2]. Moreover, bilirubin shows anti-diabetic and antilipemic properties by binding to the fat-burning nuclear receptor, peroxisome proliferator-activated receptor alpha (PPARα), to reduce body weight and blood glucose [3]. This is thought to partly explain the inverse association between circulating bilirubin concentrations and the risk of cardiovascular disease (CVD) [4,5,6,7,8], type 2 diabetes [9], rheumatoid arthritis [10], ulcerative colitis [11], lung cancer [12], and all-cause mortality [13] in adults. In children and adolescents, inverse associations between circulating bilirubin and metabolic syndrome, homeostatic model assessment of insulin resistance (HOMA-IR), glycated hemoglobin (HbA1c), and non-alcoholic fatty liver disease (NAFLD) have been reported in some [14,15,16], but not all studies [17,18].

Unconjugated bilirubin is glucuronidated in hepatocytes by the enzyme hepatic bilirubin uridine diphosphate (UDP) glucuronosyltransferase (UGT1A1) to a water-soluble form that is excreted with bile. The enzyme is encoded by the UDP glucuronosyltransferase family 1 member A1 (*UGT1A1*) gene, which is located within a gene cluster (UGT1A) on chromosome 2p37 [19]. The activity of this enzyme is regulated by the presence of a TA-repeat variant (UGT1A1*28) in the gene promoter, which is associated with Gilbert’s syndrome (benign hyperbilirubinemia). However, a mild reduction in the enzyme is not always sufficient for the full manifestation of the phenotype [20]. On the other hand, pathogenic nonsense, frameshift, missense, and splice site variants in *UGT1A1* have been reported to cause a severe reduction or complete absence of the enzyme, leading to Crigler–Najjar Syndrome (a severe form of hyperbilirubinemia) [21]. The UGT1A1*28 polymorphism is in linkage disequilibrium with single nucleotide polymorphisms (SNPs) in the UGT1A locus and associates with circulating bilirubin concentrations [22,23,24,25] and cardiometabolic diseases in adults [23,26,27,28]. A common lead SNP in this locus is rs887829, which is located 221 bp upstream of UGT1A1*28 and is in near-perfect linkage disequilibrium with UGT1A1*28 [29].

Currently, studies in children and adolescents remain limited and have mainly been performed in specific disease groups, such as patients with sickle cell anemia or neonatal hyperbilirubinemia [29,30,31]. Therefore, it remains unclear whether bilirubin is associated with cardiometabolic risk factors and inflammation in childhood and adolescence. Moreover, whether the known genetic determinants of plasma bilirubin concentrations penetrate with similar effects already during childhood or whether a different set of gene variants may play a role in determining plasma bilirubin concentrations at an early age.

In the present study, we examined relationships between fasting plasma bilirubin concentrations, cardiometabolic risk factors, and inflammation and performed a genome-wide association study (GWAS) of plasma bilirubin in Danish children and adolescents. We examined whether associations found in adults can be reproduced in children and adolescents and assessed the bilirubin heritability with respect to the UGT1A1*28-linked rs887829 polymorphism. Furthermore, we assessed the associations of rs887829 with cardiometabolic risk factors and inflammation in our cohorts.

## 2. Materials and Methods

This study was conducted in accordance with the Helsinki Declaration of 2013. All participants or their legal guardians gave written and oral informed consent, and the study was approved by the Ethics Committee of Region Zealand, Denmark (protocol no. SJ-104) and the Danish Data Protection Agency (REG-043–2013).

### 2.1. Study Population

All participants were recruited in The HOLBAEK Study (previously known as The Danish Childhood Obesity Data and Biobank) between January 2009 and September 2019. This study included 1530 unrelated Danish children and adolescents with overweight or obesity (1–19 years of age) enrolled in The Children’s Obesity Clinic (TCOC), Copenhagen University Hospital Holbæk, Holbæk, Denmark [32]. This study population will hereafter be referred to as the TCOC sample. Being overweight, including obesity, was defined as a body mass index (BMI) > 90th percentile for age and sex according to Danish BMI charts [33] (corresponding to a BMI standard deviation score (SDS) > 1.28). All measures included in the current study were obtained at the first visit to the clinic and thus prior to treatment initiation (for clinical characteristics, see Table 1). The second study population was population based and comprised 1820 unrelated Danish children and adolescents (6–19 years of age) recruited from schools in the Capital Region and Region Zeeland in Denmark.

### 2.2. Clinical Characteristics

#### 2.2.1. Medical Exclusion Criteria

In the current study, we excluded individuals (a) with known syndromes of suspected genetic origin, (b) with diseases known to affect levels of circulating bilirubin (e.g., diseases of the biliary tract or hemolytic diseases), (c) receiving pharmacological treatment that could affect bilirubin levels, liver function, or lipid homeostasis (the list included: amlodipine, captopril, desmopressin, leuprorelin, melatonin, mesalazine, montelukast, opioids, proton pump inhibitors, steroid hormones, tetracyclin, and some variants of anti-depressant, anti-epileptic, anti-psychotic, birth control, and non-steroidal anti-inflammatory drugs) [34], (d) subjected to chemotherapy and merely being cancer survivors, and (e) with a regular use of tobacco. In our study, 4% of study participants received treatment with methylphenidate, which in rare cases (<0.01%) may cause elevated plasma bilirubin [34]. When examining the levels of plasma bilirubin in our study samples, we did not observe any outliers. Furthermore, no significant difference between the bilirubin concentrations of the patients receiving methylphenidate and those of the remaining participants was observed (data not shown). Participants receiving methylphenidate treatment were therefore included in our study. 

#### 2.2.2. Questionnaire

Upon inclusion into TCOC, a pediatrician interviewed the children and adolescents, as well as their families, recording information on current or prior acute or chronic diseases or diagnoses, including known mutations or syndromes, and current use of any type of medication and smoking habits. Similar information was obtained from the control sample through self-reporting by children and adolescents and their parents through the completion of an extensive questionnaire. 

#### 2.2.3. Registration

The HOLBAEK Study, formerly known as The Danish Childhood Obesity Biobank, ClinicalTrials.gov identifier number NCT00928473, https://clinicaltrials.gov/ct2/show/NCT00928473 (registered 26 June 2009).

#### 2.2.4. Anthropometric Measures

Height was measured on a stadiometer to the nearest 0.1 cm, and weight was measured to the nearest 0.1 kg on a Tanita Digital Scale WB-110 MA (Tanita Corp., Tokyo, Japan) with the patient wearing light indoor clothes and no shoes. BMI was calculated as the weight in kilograms divided by the square of the height in meters (kg/m^2^), and BMI SDS was calculated using the LMS method [35] based on a Danish reference [32]. Waist SDS was calculated according to age- and sex-specific reference values [36]. 

#### 2.2.5. Blood Sampling and Biochemical Analyses

Following an overnight fast, venous blood samples were drawn from an antecubital vein. Total plasma bilirubin (conjugated, unconjugated, and delta-bilirubin) was quantitated using the diazo method on a Siemens Dimension Vista 11500 analyzer (Siemens AG, Munich, Germany). The coefficient of variation of the plasma bilirubin measurements decreased with increasing plasma concentrations. Within the lower (and normal physiological) plasma values, the coefficient of variation was 0.03. A proximity extension assay (https://olink.com/our-platform/our-pea-technology/, accessed on 28 April 2023) was performed to quantify inflammatory markers using the “Target 96 Cardiovascular II” and “Target 96 Inflammation” panels from Olink Proteomics AB (Uppsala, Sweden) using EDTA plasma stored at −80 °C.

Whole-body dual-energy X-ray absorptiometry was performed, and total body fat percentage was quantified in a subset from both the TCOC sample (*n* = 1168) and control sample (*n* = 73), using a GE Lunar Prodigy (DF + 10031, GE Healthcare, Madison, WI, USA). Liver fat content was measured by proton magnetic resonance spectroscopy in 458 children and adolescents. Fasting biochemical measures, previously described by our group, include plasma alanine transaminase (ALT) [37], serum high-sensitivity C-reactive protein (hs-CRP) [38], serum insulin, serum C-peptide, plasma glucose, whole blood HbA1c [39], plasma high-density lipoprotein cholesterol (HDL-C), plasma low-density lipoprotein cholesterol (LDL-C), and plasma triglycerides [40]. HOMA-IR was calculated as (insulin [mU/L] × glucose [mmol/L])/22.5 [41]. Cardiometabolic risk features, including insulin resistance, hyperglycemia, dyslipidemia, high ALT, and hypertension, were defined as previously described [42].

### 2.3. Genotyping, Imputation, and Quality Control

Extracted DNA was genotyped in-house with the Illumina Infinium Human CoreExome BeadChip (CoreExomeChip) using Illumina’s HiScan system. A total of 503,035 markers were called using Illumina Genome Studio software, and quality control was performed using Plink [43]. Individual-level quality control enforced a call-rate threshold >95% and non-outlier inbreeding coefficients (visual inspection). Outliers in a principal component analysis of ancestral markers as well as the individual having the lowest call rate of first-degree relationships were removed from the analysis. Potential sex discrepancies and sample swaps were addressed using Wunderbar [44]. We removed markers with a call rate < 95%, a minor allele frequency (MAF) < 1%, and markers with a significant deviation from Hardy–Weinberg equilibrium (HWE) (*p* < 0.001). Individuals with self-reported ethnicity other than Danish/North-European White were removed, and we included only one sibling among related individuals. 

For imputation, we used a pre-phasing approach, where haplotypes were phased without reference using SHAPEIT [45] and genotypes were imputed by IMPUTE2, applying the 1000 genomes phase 1 reference panel [46]. For subsequent analysis, we only considered imputed variants with a MAF > 1%, reasonable imputation quality *R*^2^ > 0.6, and no significant deviation from HWE (*p* > 1 × 10^−6^) after imputation, leaving ~8.5 million of the approximately 16.8 million variants in the panel for the final analyses. 

### 2.4. Genome-Wide Association Study (GWAS) 

We analyzed 8.5 million SNPs imputed from 502,718 markers passing quality control in 3350 individuals using the SNPTEST program (https://www.well.ox.ac.uk/~gav/snptest/#introduction, accessed on 28 May 2022). Genome-wide association analysis on fasting plasma concentration of bilirubin was performed in TCOC and population-based samples separately, and the results were combined in a meta-analysis using METAL (http://csg.sph.umich.edu/abecasis/Metal/, accessed on29 May 2022). 

Plasma concentrations of bilirubin were transformed using an inverse-normal rank transformation. All analyses were adjusted for age, sex, principal components, and BMI SDS. A genome-wide significance threshold of 5 × 10^−8^ was applied to the GWAS.

### 2.5. Statistical Analysis

Statistical analyses were performed in R version 4.2.0.41 ( https://www.r-project.org/, accessed on 3 April 2023). The normality of the variable distributions was evaluated. For non-parametric variables, the data were reported as the mean. Wilcoxon rank-sum tests were applied for continuous variables and χ^2^-tests for categorical variables.

For all pooled regression analyses, we constructed models with progressive adjustment. Our first regression model, Model 1, adjusted for age, sex, smoking status, and BMI SDS, except for adiposity traits, which we adjusted for age, sex, and smoking status only. Our second regression model, Model 2, included Model 1 variables plus puberty stage. Non-normally distributed (right-skewed) cardiometabolic risk factors were log-transformed. For linear regression, estimated β-effect sizes and 95% confidence intervals (CIs) were reported as the standard deviation change in cardiometabolic risk factors per SD change in bilirubin. For interaction regression analyses, we estimated whether adiposity status (overweight/obesity versus normal weight), puberty stage (pre-puberty versus puberty/post-puberty), or sex (boys versus girls) modified the associations between bilirubin and cardiometabolic risk factors/features. Statistical significance was set at *p* < 0.05, with no correction for multiple testing.

To evaluate whether variation in circulating bilirubin concentrations explained by genetics is similar between children and adults, we calculated the variance explained by the lead SNP rs887829 (heritability (*h*^2^)) using linear regression under the assumption of an additive genetic model adjusted for age, sex, and BMI SDS. 

Furthermore, we compared the effect of rs887829 in different age groups and assessed the occurrence of a potential age-dependency trend. We calculated plasma bilirubin variance and estimated the heritability of the effect of rs887829 in three age groups: Pre-pubertal (defined here as girls ≤ 8 years of age and boys ≤ 9 years of age), pubertal (defined here as girls > 8 and ≤17 years of age and boys > 9 and ≤18 years of age), and post-pubertal individuals (defined here as girls > 17 years of age and boys > 18 years of age). An interaction regression model was used to test if there was a significant difference (*p* < 0.05) in the heritability estimates of the three age groups. 

To evaluate whether the effect of the rs887829 SNP varies between the TCOC and the population-based sample and between boys and girls, interaction regression analyses were performed.

We examined the associations between bilirubin and Olink markers (cardiovascular II and Inflammation Target 96 panels). Olink batches were bridged and normalized using 16 controls using the “OlinkAnalyze” R package. Olink markers were included if more than 80% of individuals were above the limit of detection, leaving a total of 149 of the 184 proteins for analysis. The statistical significance level was set at a 5% false discovery rate (FDR) to correct for multiple testing. For all pooled regression analyses (TCOC + population-based), models were constructed with progressive adjustment. Covariates included age, sex, BMI SDS, smoking, and the year the blood sample was collected. Olink markers were inverse-normal transformed. For linear regression, estimated beta effect sizes and 95% CIs are reported as the SD change in Olink markers per SD change in bilirubin to facilitate direct comparisons of the strength of associations. Similar models were performed to test for possible associations between the SNP rs887829 and Olink markers.

## 3. Results

### 3.1. Association of Bilirubin with Cardiometabolic Risk Factors

Fasting plasma concentrations of bilirubin were inversely associated with BMI SDS and waist SDS adjusted for age, sex, and smoking status (model 1). Bilirubin was also inversely associated with hs-CRP, HOMA-IR, HbA1c, and triglycerides, adjusted for age, sex, BMI SDS, and smoking status. We observed a significantly positive association between bilirubin and plasma ALT, HDL-C, and systolic blood pressure while using model 1 (adjusted for age, sex, BMI, and smoking status). These associations persisted after adjustment for puberty stage (model 2), except for a significant inverse association for body fat percentage and LDL-C, formerly insignificant (Figure 1, Appendix A). 

### 3.2. Association of Bilirubin with Inflammatory Markers

Bilirubin (adjusted for age, sex, BMI SDS, smoking, and year of blood sample collection) was inversely associated with several plasma inflammatory markers, including CCL20, VEGFA, OPG, CTRC, IDUA, LEP, CCL11, LPL, MARCO, FABP2, CDCP1, IL18R1, AMBP, MCP2, CCL25, CXCL10, SERPINA12, CCL19, CXCL11, IFNg, MMP10, MMP12, PDL1, FGF19, IL7, MMP1, SIRT2, STK4, CCL28, AXIN1, STAMBP, SLAMF1, PRSS8, CXCL9, TNFRSF10A, IL17C, SORT1, LAP-TGFb1, IL1ra, CD40L, and IL8 and positively associated, using the same model, with GH, PTX3, THBS2, TNFRSF9, PGF, PAPPA, GT, CCL23, CX3CL1, SCF, and TRANCE (Figure 2 and Appendix A). These associations persisted after adjustment for puberty stage.

### 3.3. Genome Wide Association Study of Bilirubin

Bilirubin values were log-transformed, and the Shapiro–Wilk test was performed to examine if the variable exhibited a normal distribution. Bilirubin values were not normally distributed after the log transformation; therefore, an inverse normal rank transformation was performed. Two loci positively associated with plasma bilirubin concentrations at a *p*-value threshold of <5 × 10^−8^ (rs76999922: β = −0.65 SD; *p* = 4.3 × 10^−8^, and rs887829: β = 0.78 SD; *p* = 2.9 × 10^−247^) (Figure 3). The rs76999922 (C/A) on chromosome 8 has not yet been reported in association with fasting plasma bilirubin. The rs76999922 A allele is a bilirubin-increasing allele. There are two alternate alleles (C/T) of rs887829. The rs887829 T allele is associated with increasing fasting plasma bilirubin concentrations.

### 3.4. Heritability of rs887829 and Associations of rs887829 with Cardiometabolic Risk Factors

The narrow-sense additive heritability (*h*^2^) calculations were performed on the TCOC and control cohorts combined (*h*^2^ = 0.24; 95% CI: 0.21–0.27; *n* = 3350). When estimated in the entire control population (*n* = 1820), the narrow-sense additive heritability (*h*^2^) of the association of rs887829 with fasting plasma bilirubin concentration was *h*^2^ = 0.25 (95% CI: 0.22–0.29). When estimated in the TCOC samples (*n* = 1530), the narrow-sense additive heritability of the association of rs887829 with fasting plasma bilirubin concentration was *h*^2^ = 0.23 (95% CI: 0.18–0.27). 

No significant difference (*p* > 0.05) in heritability estimates was identified between the pre-pubertal (*h*^2^ = 0.26 [95% CI: 0.24–0.37]), pubertal (*h*^2^ = 0.24 [95% CI: 0.22–0.34]), and post-pubertal (*h*^2^ = 0.19 [95% CI: 0.10–0.24]) groups overall. We also assessed whether the heritability of the association between rs887829 and plasma bilirubin concentration differed between our two study samples. There were no significant differences in the degree of association between bilirubin and rs887829 according to BMI SDS and sex (Pinteraction > 0.05).

A linear regression in a pooled model adjusted for age, sex, smoking status, and BMI SDS (BMI SDS, waist SDS, and body fat % were not adjusted for BMI SDS) showed no significant association between rs887829 and cardiometabolic risk factors except hs-CRP and bilirubin (Figure 4, Appendix A). 

### 3.5. Association between rs887829 and Inflammatory Markers

There were no significant associations between rs887829 and the plasma inflammatory markers that were significantly associated with fasting plasma bilirubin (Appendix A). 

### 3.6. Association of Bilirubin and rs887829 with Cardiometabolic Risk Features

A 1-SD increase in bilirubin was associated with a lower prevalence of insulin resistance (odds ratio [OR] = 0.69, *p* = 2.1 × 10^−12^), hyperglycemia (OR, 0.86, *p* = 0.0096), and dyslipidemia (OR, 0.89, *p* = 0.018), but not with high ALT (OR, 1.02, *p* = 0.64), or hypertension (OR, 1.01, *p* = 0.93) (Figure 5, Appendix A). None of the cardiometabolic risk features is significantly associated with rs887829 (Figure 6, Appendix A). These associations remained consistent after adjustment for puberty stage (Figure 5 and Figure 6, Appendix A).

## 4. Discussion

In the present study, we examined the association between fasting plasma concentrations of bilirubin and cardiometabolic risk profiles and performed a GWAS of bilirubin in a sample of Danish children and adolescents with overweight or obesity (*n* = 1530) and in a population-based reference sample (*n* = 1820). Fasting plasma concentrations of bilirubin were positively associated with fasting plasma concentrations of ALT and HDL-C and systolic blood pressure (SDS). In contrast, plasma bilirubin was inversely associated with BMI SDS, waist circumference SDS, body fat percentage, hs-CRP, HOMA-IR, HbA1c, LDL-C, and triglycerides. The associations between bilirubin and cardiometabolic risk factors found in the present study are consistent with previous reports in adults [47,48,49,50]. The inverse association between bilirubin and insulin resistance might be explained to some extent since bilirubin reduces visceral obesity and insulin resistance by suppressing inflammatory cytokines and oxidative stress [49]. Recently, Kipp et al. reported that bilirubin is negatively associated with urobilin, adiposity, and insulin resistance. Urobilin, a catabolized product of bilirubin, is positively associated with insulin resistance [50]. Being an antioxidant, bilirubin protects lipids and lipoproteins from oxidation and inhibits the glycation of hemoglobin, as oxidative stress is involved in the glycation reaction [51]. 

We found significant inverse associations between plasma bilirubin and 41 circulating inflammatory markers (Figure 2). Thus, our study supports the hypothesis that a mild elevation of plasma bilirubin is associated with a higher antioxidant capacity and a lower inflammatory state in children and adolescents. Possibly due to bilirubin’s antioxidant activities and suppressive effects on pro-inflammatory cytokines, mildly elevated levels of bilirubin may protect against cardiovascular disease. Bilirubin reduces glucose and adiposity by binding to PPARα to activate the lipid-burning genes carnitine palmitoyltransferase 1 (*CPT1*) and fibroblast growth factor 21 (*FGF21*) [3]. CPT1 is a long-chain fatty acid-catalyzing enzyme, while *FGF21* encodes a hormone that is known to reduce blood glucose and adiposity [52]. Similarly, a positive association between plasma bilirubin and antioxidant capacity has been reported in adults [49].

Using GWAS, the most markedly significant association with plasma bilirubin concentrations in our GWAS analysis was identified for rs887829 located in the UGT1A1*28 locus, which has previously been strongly and positively associated with circulating bilirubin concentrations in adults [24,53,54,55]. 

Furthermore, we examined whether the variance in plasma bilirubin concentrations explained by genetics is comparable between children and adults. For this analysis, we selected rs887829, as this SNP displayed the most significant association with plasma bilirubin concentration in the GWAS analysis. rs887829 is located in the UGT1A locus, and SNPs linked to the UGT1A1*28 polymorphism within this locus have previously been reported to explain about 20% of the variation in circulating bilirubin concentrations in adults [53,54,55]. In the current study, the per-allele increasing effect on plasma bilirubin concentration for rs887829 was 0.78 SD. This effect is slightly higher than the effect previously reported in adults (0.68 SD) [54]. However, the variance in circulating bilirubin concentrations, which can be explained by rs887829 in our population (25%), was greater than the ~20% previously reported in adults for rs6742078 [54,55], which was in perfect linkage disequilibrium (*r*^2^ = 1.00, D’ = 1.00) with rs887829. We could not sufficiently explain this difference by stratifying heritability estimates into different pubertal stages. In other studies, only the point estimates of heritability are reported [54,55]. While our 95% CI for the heritability estimate lies above these previously reported point estimates, the actual heritability in the populations from which these estimates are drawn could, to some extent, deviate from the reported point estimates. 

An inverse association between the homozygote UGT1A1*28 allele with higher serum bilirubin concentrations and the risk of cardiovascular disease has been reported in some studies [27,56], but not in other studies of adults [57,58]. We tested the hypothesis of whether the UGT1A1*28-linked SNP rs887829 and higher levels of bilirubin are associated with a lower risk of cardiovascular disease in our cohort. Higher bilirubin concentrations were found to be associated with cardiometabolic risk factors in the present study. In contrast, rs887829 was not significantly associated with any of the tested cardiometabolic risk factors. Based on our findings, we hypothesize that the lower level of cardiometabolic risk factors is caused by elevated bilirubin concentrations. Likewise, previous studies in adults have found that several cardiometabolic risk factors, including BMI, circulating concentrations of LDL-C, and total cholesterol, were inversely associated with plasma bilirubin but not with the UGT1A1*28 genotype [58]. The authors observed associations between the genotype (UGT1A1*28), brachial artery diameter, and cold pressor reactivity, concluding that the observed associations of serum bilirubin levels with BMI and cholesterol were likely due to confounding and suggested that previously established cardiovascular benefits of increased plasma bilirubin may affect CVD through mechanisms associated with artery size. 

The current study involved a homogenous group of Danish/North European White children and adolescents, which is an important strength. In addition, the use of a comprehensive questionnaire enabled the careful exclusion of all individuals with diagnoses or who used drugs potentially influencing plasma bilirubin concentrations. Based on information on drug consumption, we also performed a sensitivity analysis excluding individuals receiving methylphenidate; yet, as this exclusion did not influence our findings, we report our results based on the inclusion of these individuals. Limitations to the study may include assessment at only a single time point. Future longitudinal studies are needed to validate the accuracy of the findings. The study may be subject to selection bias, with missing Tanner stage data more frequent in older participants from the obesity clinic, who are more likely to be in the pubertal or post-pubertal stage.

## 5. Conclusions

In conclusion, our study showed significant inverse associations between fasting plasma bilirubin, cardiometabolic traits, and a number of circulating inflammatory markers in Danish children and adolescents. The UGT1A1*28-linked SNP, rs887829, is associated positively with plasma bilirubin. This locus has previously been strongly associated with circulating bilirubin concentrations in adult populations. Furthermore, our findings indicate that the variance in circulating bilirubin concentrations, which can be explained by this locus, is higher in childhood compared to adulthood. The SNP rs887829 does not associate with most cardiometabolic risk factors in Danish children, which indicates that changes in bilirubin levels do not causally associate with reduced cardiometabolic risk.

## Figures and Tables

**Figure 1 antioxidants-12-01613-f001:**
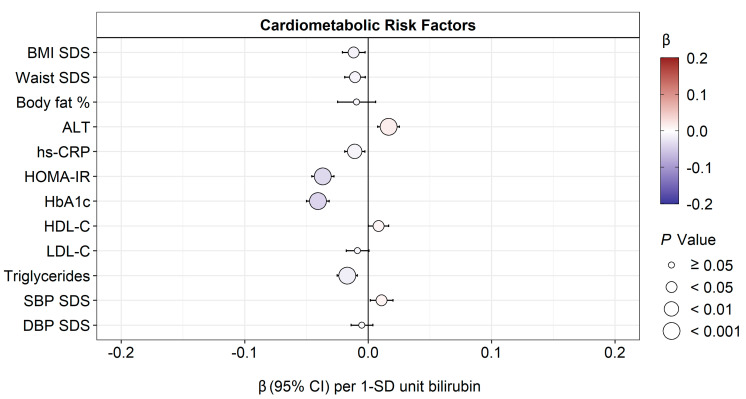
Estimated regression β-effects (95% CIs) for associations of fasting plasma bilirubin (SD units) and cardiometabolic risk factors (SD units), adjusted for age, sex, smoking status, and BMI SDS (BMI SDS, waist SDS, and body fat % were not adjusted for BMI SDS). A *p*-value < 0.05 shows statistically significant associations. Right-skewed risk factors were log-transformed.

**Figure 2 antioxidants-12-01613-f002:**
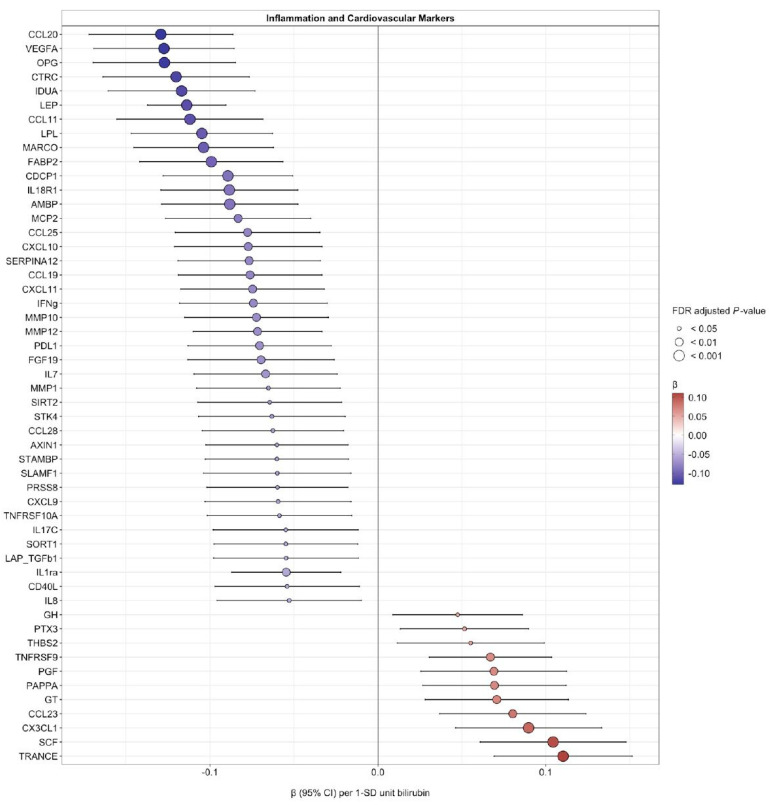
Estimated regression β-effects (95% CIs) for associations of fasting plasma bilirubin (SD-units) and plasma inflammatory markers (SD units) adjusted for age, sex, smoking status, and BMI SD score. FDR-adjusted *p*-values < 0.05 show statistically significant associations. **Abbreviations:** CCL20, C-C motif chemokine ligand 20; VEGF, vascular endothelial growth factor A; OPG, osteoprotegerin; CTRC, chymotrypsin C; IDUA, alpha-L-iduronidase; LEP, leptin; CCL11, C-C motif chemokine ligand 11; LPL, lipoprotein lipase; MARCO, macrophage receptor with collagenous structure; FABP2, fatty acid binding protein 2; CDCP1, CUB domain containing protein 1; IL18R1, interleukin 18 receptor 1; AMBP, alpha-1-microglobulin/bikunin precursor; MCP2, monocyte chemotactic protein 2; CCL25, C-C motif chemokine ligand 25; CXCL10, C-X-C motif chemokine ligand 10; SERPINA12, serpin family A member 12; CCL19, C-C motif chemokine ligand 19; CXCL11, C-X-C motif chemokine ligand 11; IFNg, interferon gamma; MMP10, matrix metallopeptidase 10; MMP12, matrix metallopeptidase 12; PDL1, programmed cell death 1 ligand 1; FGF19, fibroblast growth factor 19; IL7, interleukin 7; MMP1, matrix metallopeptidase 1; SIRT2, sirtuin 2; STK4, serine/threonine kinase 4; CCL28, C-C motif chemokine ligand 28; AXIN1, axin 1; STAMBP, STAM binding protein; SLAMF1, signaling lymphocytic activation molecule family member 1; PRSS8, serine protease 8; CXCL9, C-X-C motif chemokine ligand 9; TNFRSF10A, TNF receptor superfamily member 10a; IL17C, interleukin 17C; SORT1, sortilin 1; LAP_TGFb1, latency-associated peptide transforming growth factor beta-1; IL1ra, interleukin-1 receptor antagonist; CD40L, cluster of differentiation 40 ligand; IL8, interleukin 8; GH, growth hormone; PTX3, pentraxin 3; THBS2, thrombospondin 2; TNFRSF9, TNF receptor superfamily member 9; PGF, placental growth factor; PAPPA, pappalysin 1; GT, gastrotropin; CCL23, C-C motif chemokine ligand 23; CX3CL1, C-X3-C motif chemokine ligand 1; SCF, stem cell factor; TRANCE, TNF-related activation-induced cytokine.

**Figure 3 antioxidants-12-01613-f003:**
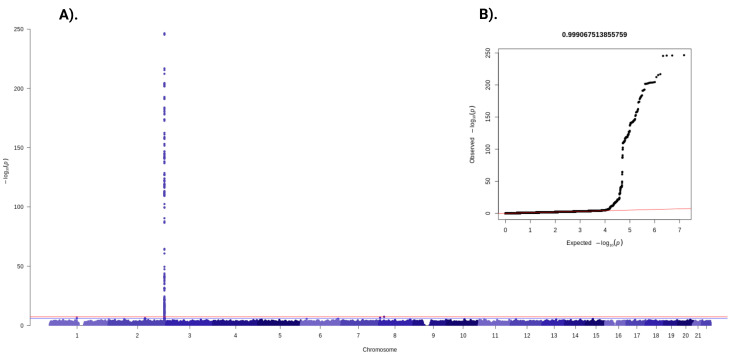
(**A**,**B**) Manhattan plot and Q–Q plot of genome-wide SNPs for plasma bilirubin in Danish children and adolescents. SNPs are plotted on the x-axis according to their chromosomal position against the –log_10_ (*p*-value). Associations with a *p*-value ≤ 5 × 10^−8^ were considered genome-wide significant (red line), and a *p*-value ≤ 1 × 10^−6^ is marked as genome-wide suggestive (blue line).

**Figure 4 antioxidants-12-01613-f004:**
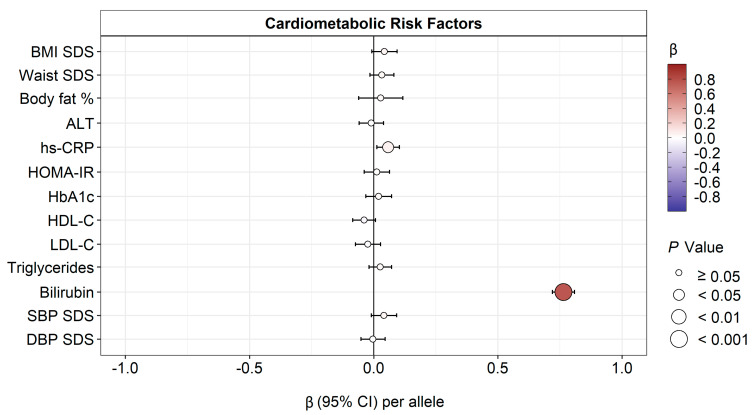
Associations of bilirubin-increasing rs887829 allele with cardiometabolic risk factors. No significant association was found between rs887829 and cardiometabolic risk factors except hs-CRP and bilirubin. **Abbreviations**: BMI, body mass index; ALT, alanine aminotransferase; hs-CRP, high-sensitivity C-reactive protein; HOMA-IR, homeostasis model assessment of insulin resistance; HbA1c, hemoglobin A1c; HDL-C, high-density lipoprotein cholesterol; LDL-C, low-density lipoprotein cholesterol; SBP, systolic blood pressure; DBP, diastolic blood pressure; SDS, SD score.

**Figure 5 antioxidants-12-01613-f005:**
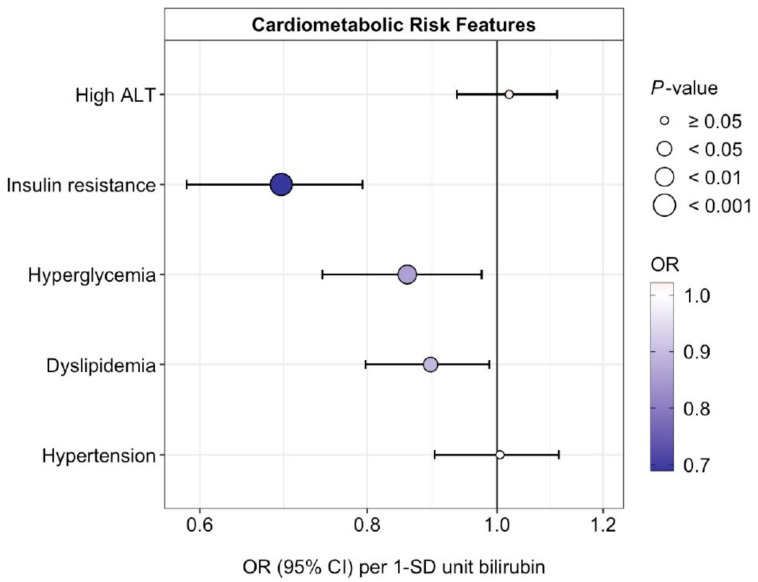
Estimated odds ratios (ORs) (95% CIs) for associations of fasting plasma bilirubin (SD-units) and cardiometabolic risk features, adjusted for age, sex, smoking status, and BMI SDS. **Abbreviations:** ALT, alanine transaminase (surrogate measure of hepatic steatosis); SDS, SD score.

**Figure 6 antioxidants-12-01613-f006:**
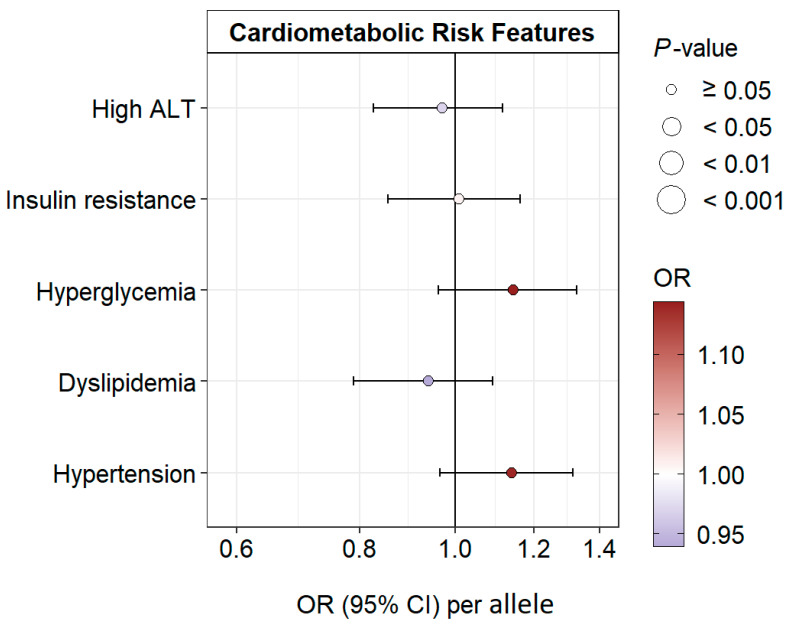
Estimated odds ratios (OR) (95% CIs) for associations of bilirubin-increasing rs887829 T allele and cardiometabolic risk features, adjusted for age, sex, smoking status, and BMI SDS.

**Table 1 antioxidants-12-01613-t001:** Clinical characteristics of study participants.

	Overall	Population	TCOC	*p*
N	3350	1820	1530	
Age (mean (SD)), year	11.86 (3.28)	12.00 (3.55)	11.68 (2.92)	0.005
Height (mean (SD)), cm	152.71 (17.04)	152.13 (18.32)	153.41 (15.35)	0.030
Plasma bilirubin (mean (SD)), µmol/L	7.93 (3.73)	8.05 (3.75)	7.78 (3.69)	0.041
BMI SDS (mean (SD)),	1.47 (1.57)	0.29 (1.03)	2.89 (0.67)	<0.001
Waist (mean (SD)), cm	78.53 (17.33)	67.10 (9.34)	91.95 (14.74)	<0.001
Hip (mean (SD)), cm	88.73 (15.29)	82.10 (12.39)	96.53 (14.67)	<0.001
Total fat mass (mean (SD)), %	27.79 (12.44)	11.58 (5.01)	28.85 (12.04)	<0.001
HDL_chol (mean (SD)), mmol/L	1.39 (0.34)	1.51 (0.33)	1.24 (0.30)	<0.001
LDL_chol (mean (SD)), mmol/L	2.28 (0.67)	2.12 (0.60)	2.47 (0.70)	<0.001
Total_chol (mean (SD)), mmol/L	4.05 (0.74)	3.94 (0.68)	4.18 (0.79)	<0.001
Triglycerides (mean (SD)), mmol/L	0.85 (0.48)	0.68 (0.31)	1.06 (0.57)	<0.001
Insulin (mean (SD)) mIU/L	74.85 (62.61)	56.56 (30.40)	96.61 (81.32)	<0.001
HOMA-IR (mean (SD)), mIU/L	3.14 (3.08)	1.92 (1.15)	3.75 (3.54)	<0.001
HbA1c (mean (SD)), mmol/mol	33.85 (3.42)	33.63 (3.33)	34.10 (3.49)	<0.001
Glucose (mean (SD)), mmol/L	5.06 (0.53)	4.97 (0.40)	5.10 (0.57)	<0.001
C-peptide (mean (SD)), pmol/L	612.81 (287.25)	503.91 (187.17)	756.86 (330.15)	<0.001
ALT (mean (SD)), U/L	24.04 (13.57)	20.71 (7.34)	28.03 (17.65)	<0.001
ASAT (mean (SD)), U/L	27.27 (13.26)	27.33 (14.57)	27.19 (11.30)	0.782
hs-CRP (mean (SD)), mg/dL	2.39 (6.74)	1.43 (5.84)	3.54 (7.53)	<0.001
Liver fat % (mean (SD))	2.98 (6.03)	0.65 (0.34)	3.15 (6.21)	0.042

BMI: body mass index; SD: standard deviation; TCOC: The Children’s Obesity Clinic; HDL_chol: high-density lipoprotein cholesterol; LDL_chol: low-density lipoprotein cholesterol; HOMA-IR: homeostatic model assessment for insulin resistance; HbA1c: hemoglobin A1c; ALT: alaninaminotransferase; ASAT: aspartate transaminase; hsCRP: high-sensitivity C-reactive protein.

## Data Availability

The data in this study are from The HOLBAEK Study, a part of the research activities in TARGET (www.target.ku.dk) and BIOCHILD (www.biochild.ku.dk). The ethical approval limits individual-level data availability and prohibits the authors from making the minimal data set publicly available. Data are available from the corresponding author (Torben Hansen) upon ethical approval from the Regional Ethical Committee of Region Zealand and require a data processing agreement between the researcher and The Danish Childhood Obesity Biobank. The steering committee of The HOLBAEK Study can be contacted via the corresponding authors (Torben Hansen, torben.hansen@sund.ku.dk and Jens-Christian Holm, jhom@regionsjaelland.dk). Any publications arising from the use of this dataset must acknowledge the source of the dataset, its funding, and the collaborative group that collected the data. Data is safely stored in a database with a central backup.

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
