# Peer review of "Genetics of Plasma Bilirubin and Associations between Bilirubin and Cardiometabolic Risk Profiles in Danish Children and Adolescents"

_antioxidants, 2023, doi:10.3390/antiox12081613_

Round 1

Reviewer 1 Report

There are some concerns below that can improve the manuscript:

·      Abstract: The authors wrote in the abstract, “Several genetic loci associating with circulating bilirubin concentrations have been identified, genome-wide association studies in adults.” The statement ‘several’ is not correct, there are some, but several is a stretch. Please rewrite it to be correct.

·      Abstract: The first 3 sentences of the abstract could be blended not to be redundant as it currently reads. Please give the whole manuscript a close read throughout as this occurs in other areas too.

·      Abstract: In the abstract, the sentence “The GWAS showed that UGT1A, rs887829 associated strongly and positively with plasma bilirubin (β = 0.78 SD, P = 2.9⸱10-247). Approximately 25% of the variance in plasma bilirubin concentration was explained by rs887829.” It could be earlier in the abstract. Please rewrite the abstract to have a better flow for the readers.

·      Introduction: In the second paragraph of the abstract, the authors mention only UGT1A1*28 but mention ‘several genetic loci’ in the abstract. Please add more here if it is relevant to discuss.

·      Table 1: The BMIs of the kids do not seem to be correct. Please explain if it is correct.

·      Results: In the first paragraph of the results, the authors write “and smoking status” –this study was on children that are around 12 years old; how many of these children smoke? This seems to be a variable that is not reliable and possibly not needed. Please explain.

·      Figures: All of the figures need descriptive titles and a better explanation in the figure legend.

·      Discussion: In the discussion section, the authors wrote, “inverse association between bilirubin and insulin resistance might be explained to some extent, since bilirubin reduces visceral obesity and insulin resistance by suppression of inflammatory cytokines and oxidative stress [48].” The manuscript that they reference is on atherosclerosis and not insulin resistance. There is current work that would better support these claims such as by Kipp et al. that showed that bilirubin is negatively associated with adiposity and insulin resistance and that the bilirubin breakdown product, urobilin, is positively associated with insulin resistance. The authors should expand upon the discussion of the role of bilirubin in regulating BMI and insulin resistance.

·      Discussion: In the second paragraph of the discussion, the authors wrote, “Thus, our study supports the hypothesis that a mild elevation of plasma bilirubin is associated with a higher antioxidant capacity and a lower inflammatory state in children and adolescents.” The authors do not discuss the latest findings on bilirubin and that it functions as a hormone to burn fat. The authors should discuss this aspect in the introduction and discussion as it is very relevant to the work performed and being discussed.

Author Response

Reviewer 1

Comments and Suggestions for Authors

There are some concerns below that can improve the manuscript:

Abstract: The authors wrote in the abstract, “Several genetic loci associating with circulating bilirubin concentrations have been identified, genome-wide association studies in adults.” The statement ‘several’ is not correct, there are some, but several is a stretch. Please rewrite it to be correct.

Response: Thank you for the suggestion. We have made the suggested changes in the revised manuscript. The changes we have made in the main text are highlighted in yellow.

The following changes have been made.

Line 17-19; “Some genetic loci associating with circulating bilirubin concentrations have been identified by genome-wide association studies in adults”.

Abstract: The first 3 sentences of the abstract could be blended not to be redundant as it currently reads. Please give the whole manuscript a close read throughout as this occurs in other areas too.

Response: We have rephrased the mentioned sentences. In the revised manuscript, we have made several changes and we believe that it would have improved the article readability.  The following changes have been made.

Line 14-18; “Bilirubin is the end product of the heme catabolism mainly produced by the breakdown of mature red blood cells. Due to its anti-inflamatory, antioxidant, antidiabetic, and antilipemic properties, circulating bilirubin concentrations are inversely associated with the risk of cardiovascular disease, type 2 diabetes, and all-cause mortality in adults. Some genetic loci associating with circulating bilirubin concentrations have been identified by genome-wide association studies in adults.

Abstract: In the abstract, the sentence “The GWAS showed that UGT1A, rs887829 associated strongly and positively with plasma bilirubin (β = 0.78 SD, P = 2.9⸱10-247). Approximately 25% of the variance in plasma bilirubin concentration was explained by rs887829.” It could be earlier in the abstract. Please rewrite the abstract to have a better flow for the readers.

Response: Thank you for the comment. We have considered the point raised by the worthy reviewer and we appreciate their suggestion, but we believe that the sequence of findings of the present study are according to the analysis plan. As in the present study, first we have tested the association between bilirubin, inflammatory cytokines and cardiometabolic risk factors. Then, GWAS was performed to examine genetic architecture of plasma bilirubin concentrations in our cohort. Therefore, first, we have discussed association of bilirubin with cardiometabolic factors and then the findings of GWAS.  We have used the same sequence of research plan in methods, results and discussion sections. Therefore, we believe that keeping the same sequence to add findings of the study in abstract, methods, results and discussion sections will have a better flow for the readers.

Introduction: In the second paragraph of the abstract, the authors mention only UGT1A1*28 but mention ‘several genetic loci’ in the abstract. Please add more here if it is relevant to discuss.

Response: In literature, several studies have been performed to examine genetic influence on circulating bilirubin in adults. Several genetic loci associating with bilirubin have been reported. Among these loci, the most important locus associating with bilirubin is UGT1A locus due to the UGT1A1 gene encoding UGT1A1 enzyme that converts bilirubin to water-soluble form. The UGT1A1 locus, and SNPs linked to the UGT1A1*28 polymorphism within this locus have previously been reported to explain about 20% of the variation in circulating bilirubin concentrations in adults. In the present study, The GWAS showed that two loci positively associated with plasma bilirubin concentrations at P value threshold of < 5 x 10-8 (rs76999922: β = -0.65 SD; P = 4.3e-8, and rs887829: β = 0.78 SD; P = 2.9e-247).  We found UGT1A, rs887829 associating strongly and positively with plasma bilirubin explaining 25% of the variance in plasma bilirubin concentration. Therefore, we have only focused on UGT1A1 locus and UGT1A1*28-linked SNP, rs887829 in our introduction.  

The following changes have been made.

 Line 35-37; “The GWAS showed that two loci positively associated with plasma bilirubin concentrations at P value threshold of < 5 x 10-8 (rs76999922: β = -0.65 SD; P = 4.3e-8, and rs887829: β = 0.78 SD; P = 2.9e-247).”

Table 1: The BMIs of the kids do not seem to be correct. Please explain if it is correct.

Response: The table shows BMI SDS not BMIs of the kids. A threshold of BMI SDS greater than 1.28 was used to define overweight including obesity.

Results: In the first paragraph of the results, the authors write “and smoking status” –this study was on children that are around 12 years old; how many of these children smoke? This seems to be a variable that is not reliable and possibly not needed. Please explain.

Response: The participants of the present study have ages 1-18 years with mean age 12 years. There were 124 active smokers in the cohorts that were excluded. We did not exclude participants based on passive smoking and used it as a covariate. It has been reported that smoking cessation is followed by increases in bilirubin concentration that have been associated with lower risk of lung cancer and cardiovascular disease (PMID: 24812024).

Figures: All of the figures need descriptive titles and a better explanation in the figure legend.

Response: Thank you for the suggestion. We have added descriptive titles and a better explanation to the figure legend in the revised version of the manuscript.

Discussion: In the discussion section, the authors wrote, “inverse association between bilirubin and insulin resistance might be explained to some extent, since bilirubin reduces visceral obesity and insulin resistance by suppression of inflammatory cytokines and oxidative stress [48].” The manuscript that they reference is on atherosclerosis and not insulin resistance. There is current work that would better support these claims such as by Kipp et al. that showed that bilirubin is negatively associated with adiposity and insulin resistance and that the bilirubin breakdown product, urobilin, is positively associated with insulin resistance. The authors should expand upon the discussion of the role of bilirubin in regulating BMI and insulin resistance.

Response: Thank you for the helpful comment. We have corrected the reference and expanded the discussion by adding the association of bilirubin and urobilin with adiposity and insulin resistance.

The following changes have been made in the revised manuscript.

Line 367-369; “Recently, Kipp et al. reported that bilirubin associated negatively with urobilin, adi-posity, and insulin resistance. Urobilin, a catabolized product of bilirubin is positively associated with insulin resistance [50].”

Discussion: In the second paragraph of the discussion, the authors wrote, “Thus, our study supports the hypothesis that a mild elevation of plasma bilirubin is associated with a higher antioxidant capacity and a lower inflammatory state in children and adolescents.” The authors do not discuss the latest findings on bilirubin and that it functions as a hormone to burn fat. The authors should discuss this aspect in the introduction and discussion as it is very relevant to the work performed and being discussed.

Response: The role of bilirubin as a ligand to bind to lipid burning receptor PPAR alpha has been added in the introduction and discussion section of the revised manuscript.

The following paragraphs were added in introduction and discussion section.

Line 47-50; “Moreover, bilirubin shows anti-diabetic and antilipemic properties by binding to the fat burning nuclear receptor, peroxisome proliferator-activated receptor alpha (PPARα) to reduce body weight and blood glucose [3].”

Line 378-381; “Bilirubin reduces glucose and adiposity by binding to PPARα to activate the lipid burning genes carnitine palmitoyltransferase 1 (CPT1) and fibroblast growth factor 21 (FGF21) [3]. CPT1 is a long-chain fatty acids catalyzing enzyme while, FGF21 encodes a hormone, which is known to reduce blood glucose and adiposity [52].”

Reviewer 2 Report

This article describes the association between fasting plasma concentrations of bilirubin on cardiometabolic risk profiles and performed a GWAS of bilirubin in a sample of Danish children and adolescents with overweight or obesity and in a population-based reference sample. The authors found significant inverse associations between plasma bilirubin and 41 circulating inflammatory cytokines, but no associated between UGT1A1*28-linked SNP, rs887829 with any of the mentioned cardiometabolic risk factors except for hs-CRP and the correlation was age dependent. The authors conclude that plasma concentrations of bilirubin non-causally associates with cardiometabolic risk factors in  children and adolescents. The paper is well written, and the clinical study was well prepared and performed. I have only minor comments listed below:

Abstract is clear; however the authors write that the level of bilirubin correlated with “cytokines GH, PTX3 (…)”. There must be a mistake as growth hormone (GH) is not a cytokine. The authors should verify it.

Introduction is informative and introduce the reader to the topic, as well as explain the aim of the study.

Material and methods- there are all necessary information regarding patients’ recruitment, ethical committee approval, samples collection, performed biochemical analysis and statistical analysis.

Results- Clear. The fond size on Figure 3 could be bigger.

The authors should add limitations to the study at the end of discussion.

Author Response

Reviewer 2:

This article describes the association between fasting plasma concentrations of bilirubin on cardiometabolic risk profiles and performed a GWAS of bilirubin in a sample of Danish children and adolescents with overweight or obesity and in a population-based reference sample. The authors found significant inverse associations between plasma bilirubin and 41 circulating inflammatory cytokines, but no associated between UGT1A1*28-linked SNP, rs887829 with any of the mentioned cardiometabolic risk factors except for hs-CRP and the correlation was age dependent. The authors conclude that plasma concentrations of bilirubin non-causally associates with cardiometabolic risk factors in children and adolescents. The paper is well written, and the clinical study was well prepared and performed. I have only minor comments listed below:

Abstract is clear; however, the authors write that the level of bilirubin correlated with “cytokines GH, PTX3 (…)”. There must be a mistake as growth hormone (GH) is not a cytokine. The authors should verify it.

Response: We thank the worthy reviewer for highlighting this important point. We examined the associations between bilirubin and Olink markers (cardiovascular II and Inflammation Target 96 panels (https://olink.com/products-services/target/48-cytokine-panel/)). GH is included in the Olink cardiovascular panel (1024-v1.3-cvd-ii-panel-content-final.pdf (olink.com)). It plays an important role in growth control. Its major role in stimulating body growth is to stimulate the liver and other tissues to secrete IGF-1. It stimulates both the differentiation and proliferation of myoblasts. It also stimulates amino acid uptake and protein synthesis in muscle and other tissues (Protein - Olink). We have changed the phrase “inflammatory cytokine” to “inflammatory markers”.

We have made the following correction in the revised manuscript.

Line 35-37; “In contrast, bilirubin was positively associated with fasting plasma concentrations of alanine transaminase (ALT), high-density lipoprotein cholesterol (HDL-C), and systolic blood pressure SDS, and the inflammatory markers GH, PTX3, THBS2, TNFRSF9, PGF, PAPPA, GT, CCL23, CX3CL1, SCF, and TRANCE.”

Introduction is informative and introduce the reader to the topic, as well as explain the aim of the study.

Material and methods- there are all necessary information regarding patients’ recruitment, ethical committee approval, samples collection, performed biochemical analysis and statistical analysis.

Response: We highly appreciate positive comments by the worthy reviewer.

Results- Clear. The fond size on Figure 3 could be bigger.

Response: A revised figure with bigger font size is added in the revised version.

The authors should add limitations to the study at the end of discussion.

Response: We thank the worthy reviewer for pointing this out to us. We have added the limitations to the study at the end of discussion in the revised manuscript.

The following changes have been made.

Line 428-432; “Limitations to the study may include assessment at only a single time point. Future longitudinal studies are needed to validate accuracy of the findings. . The study may be subject to selection bias with missing Tanner stage data more frequent in older participants from the obesity clinic, who are more likely to be in the pubertal or post-pubertal stage.”